# Spiradenoma: A Case Report and Review of the Literature

**DOI:** 10.3390/diagnostics15020173

**Published:** 2025-01-14

**Authors:** Jia-Ying Chang, Yen-Chang Chen, Dah-Ching Ding

**Affiliations:** 1Department of Obstetrics and Gynecology, Hualien Tzu Chi Hospital, Buddhist Tzu Chi Medical Foundation, Tzu Chi University, Hualien 970, Taiwan; julia100100100@gmail.com; 2Division of Digital Pathology, Department of Anatomical Pathology, Hualien Tzu Chi Hospital, Buddhist Tzu Chi Medical Foundation, Hualien 970, Taiwan; s92312129@gmail.com; 3Department of Pathology, School of Medicine, Tzu Chi University, Hualien 970, Taiwan; 4Institute of Medical Sciences, Tzu Chi University, Hualien 970, Taiwan

**Keywords:** spiradenoma, skin adnexa, subcutaneous, CK7, surgical excision

## Abstract

**Background and Clinical Significance:** Spiradenoma is a rare benign skin adnexal tumor with unknown incidence and prevalence, typically affecting young to middle-aged adults without a sexual predilection. **Case Presentation:** A 59-year-old woman presented with a palpable lesion in the suprapubic region that had been there for 20 years and had become enlarged over the past 2 months. Physical examination revealed a firm, non-tender, subcutaneous mass, approximately 2 cm in size, in the right pubic region. Ultrasound revealed a hypoechoic, heterogeneous lesion with a well-defined border, measuring 2.37 × 0.94 × 1.67 cm, without hypervascularity. Therefore, the patient underwent excision of the subcutaneous tumor. The pathology report confirmed the diagnosis of spiradenoma of the pubis. Histochemistry showed that the inner luminal cells were positive for CK7, and the outer basaloid cells were positive for p63. CD56 and CD117 were focally positive. **Conclusions:** With an accurate diagnosis and appropriate surgical excision, the prognosis for spiradenoma is generally excellent. However, a long-term follow-up is advisable.

## 1. Introduction

Spiradenoma is a rare benign skin adnexal tumor (SAT) with unknown incidence and prevalence, typically affecting young to middle-aged adults without a sexual predilection [1,2]. It usually presents as a solitary nodule, often on the scalp, and can be challenging to diagnose clinically and dermoscopically because of its nonspecific appearance [1,2].

According to D’Andrea et al., among 281 SAT cases diagnosed within 20 years, spiradenoma was the most frequent benign SAT (19.5%), with a slight male predominance [3]. Multiple segmental forms have been reported [3]. Malignant transformation to spiradenocarcinoma is rare but possible, especially in long-standing lesions in elderly patients [2,4]. Morphologically, low-grade spiradenocarcinomas tend to have an indolent course, with a local recurrence rate of 19%, but no metastases or disease-related mortality [4]. Therefore, complete surgical excision is the recommended treatment [2].

Here, we report the case of a woman who presented with a subcutaneous tumor that was diagnosed as spiradenoma. The patient provided consent for the publication of this case report.

## 2. Case Report

A 59-year-old woman presented with a palpable suprapubic lesion persisting for 20 years, which had enlarged over the previous 2 months. Her medical history included systemic lupus erythematosus (SLE) with nephritis diagnosed at age 20, managed with hydroxychloroquine until 2021; osteoporosis treated with denosumab injections; and a nodular thyroid goiter following a left thyroid lobectomy in 2015. She was also prescribed estradiol valerate (1 mg) and medroxyprogesterone acetate (2.5 mg) daily from 2019 to 2021 for postmenopausal syndrome and thyroxine for hypothyroidism.

The patient had no significant personal or family history. Physical examination revealed a firm, non-tender, subcutaneous mass approximately 2 cm in size in the right pubic region. Laboratory results, including complete blood count, renal function, liver enzymes, and electrolytes, were normal. Ultrasound demonstrated a hypoechoic, heterogeneous lesion with well-defined borders, measuring 2.37 × 0.94 × 1.67 cm, without evidence of hypervascularity (Figure 1).

The patient underwent surgical excision of the subcutaneous tumor. Under general anesthesia with a laryngeal mask, the patient was positioned in lithotomy. A 2 cm linear incision was made over the pubic tumor, extending into the underlying tissue. The tumor was excised intact with a 1 cm safe margin (Figure 2). The wound was closed with subcutaneous sutures and dressed with gauze. The procedure was uneventful, with no surgical complications.

The pathology report confirmed the diagnosis of spiradenoma in the pubic region. Microscopically, the tumor was identified as a cutaneous adnexal neoplasm with a well-defined multilobular architecture. It comprised outer basaloid cells displaying a basal cell or myoepithelial phenotype and inner luminal cells arranged in solid, trabecular, and tubular patterns, with areas of focal chondromyxoid stroma (Figure 3).

Immunohistochemical staining was positive for pan-cytokeratin in both cell types. CK7 was expressed in the inner luminal cells, while p63 was positive in the outer basaloid cells. Focal positivity was observed for CD56 and CD117, whereas PAX8 was negative. The Ki67 proliferation index ranged from approximately 10% to 20% (Figure 4).

## 3. Literature Review and Discussion

### 3.1. Search Strategy

A systematic search was conducted using the keywords “spiradenoma, skin, and adult” from 2010 to 10 December 2024. Synonyms and related terms were also included to expand the scope. The bibliographies of relevant reviews and included studies were also examined. Table 1 provides an overview of the search strategy used for the PubMed, Scopus, Web of Science, and Embase databases.

### 3.2. Definition, Epidemiology, and Demographics

Eccrine spiradenoma is a rare, benign adnexal tumor originating from the sweat glands and is typically located in the dermal or subcutaneous layers [5]. It can occur anywhere on the body; however, the limbs remain the most affected site [3]. In our patient, it was located in the pubis region.

Cure et al. reported that Brooke–Spiegler syndrome (BSS) is a rare autosomal dominant disorder marked by the development of multiple adnexal tumors, primarily trichoepitheliomas, cylindromas, and occasionally spiradenomas. These lesions typically emerge during the second or third decade of life, with a potential risk of malignant transformation in pre-existing tumors [6].

D’Andrea et al. collected 281 SAT cases during a 20-year period [3]. They found that 94.3% of cases are benign, with the most common histological types being eccrine spiradenoma, hidrocystoma, eccrine poroma, syringoma, sebaceous adenoma, and trichofolliculoma. SATs are more common in men than in women. The mean age of the patients was 59 years. Recurrence is rare. Malignant SATs account for 5.9% of all cases. The authors concluded that, overall, SATs are rare, and the majority are benign. Accurate diagnosis and thorough surgical excision are crucial [3].

### 3.3. Etiology and Pathogenesis

Eccrine spiradenoma, a tumor of the sweat glands, may appear congenitally or spontaneously, with an uncertain etiology [7,8]. BSS is a rare autosomal dominant disorder that may also be a factor in the etiology of spiradenoma [9]. Prior trauma is considered a contributing factor [10]. A multipotent stem cell origin is also one of the theories for the etiology [8].

### 3.4. Clinical Features (Typical Presentation, Common Location on the Body, and Differential Diagnosis)

A spiradenoma typically presents as a tender, slow-growing nodule on the trunk or extremities [11]. Solitary familial cases of autosomal dominant inheritance have also been reported [12]. The most common symptom is the accelerated growth of a longstanding lesion [13]. Our patient had no symptoms for 20 years, followed by increased tumor growth rate.

D’Andrea et al. reported locations of spiradenoma on the limbs (n = 17), back (n = 14), torso (n = 9), scalp (n = 7), face (n = 4), skin (n = 2), and periocular region (n = 2) [3]. The breast was also reported as a rare site for spiradenoma [14,15]. The pubic region located in the lower torso, as in our case, is a rare tumor location.

The differential diagnosis of eccrine spiradenoma can be challenging due to its rarity and occasional coexistence with other lesions. Conditions to consider include cylindroma, hidradenoma, and malignant tumors such as spiradenocarcinoma and adenoid cystic carcinoma [15]. Cylindroma is morphologically distinguished from eccrine spiradenoma by its characteristic puzzle-like cell arrangement within non-encapsulated dermal nodules; its pathological features include PAS positivity and hyalinized basement membrane material [15]. In spiradenoma, none of these features are observed. Hidradenoma typically comprises uniform polygonal cells that may occasionally exhibit squamoid features. It is characterized by the expression of carcinoembryonic antigen, epithelial membrane antigen, and GCDFP-15 [15].

Histologically, spiradenoma is characterized by two cell populations: small dark basaloid cells and large pale cells [16]. The tumor typically presents as a well-demarcated and lobulated intradermal tumor with one or more lobules located in the dermis, tightly packed small and large groups of cells, or arranged in diffuse alveolar or pseudorosette formations [15]. Histopathological examination is often required for a definitive diagnosis, and immunohistochemical analysis shows positive staining for CK5/CK6, CK8/CK18, and S100 [17].

Immunohistochemical staining can help identify specific cell types within the tumor; small cells express p63 and calponin, and large cells express CK7 and CD117 [15]. Adenoid cystic carcinoma (ACC)-like components consist of epithelial basaloid cells surrounding pseudocystic structures filled with mucinous material [15]. Understanding these histological characteristics is crucial for an accurate diagnosis and differentiation from other cutaneous neoplasms. The previous study showed that the lesion was negative for estrogen receptor, progesterone receptor, androgen receptor, and HER2 and exhibited a low proliferation index (<5%) [15].

Our case findings are comparable to those of previous studies. The inner luminal cells were positive for CK7, and the outer basaloid cells were positive for p63. CD56 and CD117 were focally positive. The Ki67 index was approximately 10–20%.

The diagnosis of spiradenoma can be challenging because of its nonspecific clinical appearance and dermoscopic features, which may mimic other cutaneous tumors [1]. Imaging studies, including magnetic resonance imaging (MRI) and ultrasound, can aid in the diagnosis of well-circumscribed lesions with characteristic signal intensities [18,19]. Lee et al. reported the MRI characteristics of eccrine spiradenoma in the breast, where the tumor exhibited low signal intensity on T1-weighted images, intermediate to low signal intensity on T2-weighted images, and homogeneous enhancement on T1-weighted contrast-enhanced images [20]. Epidermal cysts appear as well-defined masses with variable echogenicity on ultrasound, reflecting their diverse composition, and they typically lack internal vascularity [5]. Jin et al. described ultrasound findings for eccrine spiradenoma in the upper arm, presenting as a well-defined, lobulated mass with heterogeneous hypoechogenicity located in the deep dermis and superficial subcutaneous fat layer, without epidermal connection or extension into muscular structures [19].

Dermoscopy typically revealed a homogenous pink area with arborizing telangiectasias and occasional blue-ovoid nests or globules [1].

Imaging techniques, such as computed tomography (CT) and positron-emission tomography (PET)/CT, are recommended for staging and detecting metastases in suspected malignant cases [13]. Petitto et al. reported that a PET/CT scan of a malignant spiradenoma revealed a fluorodeoxyglucose (FDG)-avid mass with an SUVmax of 22.9, along with FDG-avid right external iliac lymph nodes showing SUVmax values of 9.8 and 12 and left inguinal lymph nodes with SUVmax values of 5.2 and 5.1 [21].

Diagnosis can be challenging because of its nonspecific clinical appearance and dermoscopic features, which may mimic other cutaneous tumors [1]. Diagnosis often requires a biopsy because the clinical presentation can vary [11]. Aspiration cytology can be useful in diagnosing adnexal tumors, including spiradenoma, with a reported accuracy rate of 83.3% in one study [22]. Lee et al. reported using ultrasound-guided core needle biopsy for pathological examination, which revealed spiradenoma [23]. However, cytology is limited for assessing local invasion, which is crucial for distinguishing benign eccrine spiradenoma from its malignant counterpart, spiradenocarcinoma [22]. A proper diagnosis is important because of the rare possibility of malignant transformation [1,11].

Our patient underwent an ultrasound examination, which revealed a subcutaneous well-defined border tumor with little blood flow.

### 3.5. Treatment and Management

Surgical excision is the gold-standard treatment for eccrine spiradenoma with low recurrence rates [7]. Radical excision is widely accepted as the primary treatment [24,25]. The treatment typically involves surgical excision with CO_2_ laser ablation, which is an additional option for benign cases [12,26]. Clear resection margins and tumor-free regional lymph nodes are crucial for reducing recurrence and mortality in malignant cases [13]. However, Mohs micrographic surgery has been successfully used in at least one case of spiradenocarcinoma [24]. The prognosis of malignant spiradenomas is unpredictable, and it should be considered potentially lethal [25]. Consistent with previous studies, our patient underwent complete surgical excision.

### 3.6. Prognosis and Follow-Up

Although the prognosis is generally excellent, rare cases of malignant transformation have been reported [1,7]. Although rare, malignant transformations can occur and may behave aggressively [27]. Recent genomic analysis has revealed mutations in the CYLD gene in some cases, as well as a recurrent missense mutation in the ALPK1 gene, which can activate the NF-κB pathway [28].

Regular follow-up is crucial, as demonstrated in a case with no recurrence 2 years after excision of only the malignant area [25]. In one reported case, the patient remained disease-free during a 1-year follow-up period after complete tumor excision [29]. Although spiradenoma is generally benign, in rare cases, malignant transformation can occur after an extended latency period [30]. This underscores the importance of a proper diagnosis and follow-up.

### 3.7. Future Direction and Research Needs

#### 3.7.1. Understanding Pathogenesis

Further research is necessary to uncover the molecular and genetic mechanisms underlying the development of spiradenoma. Identifying key genetic mutations, such as those associated with BSS, and exploring pathways involved in tumor growth and progression could provide insights for targeted therapies [31].

#### 3.7.2. Improved Diagnostic Approaches

Developing advanced diagnostic tools, such as molecular markers or imaging techniques, could aid in distinguishing spiradenoma from similar adnexal tumors such as cylindroma and hidradenoma. Additionally, creating more robust criteria for diagnosing rare variants and malignant transformations (e.g., spiradenocarcinoma) would enhance clinical precision [32].

#### 3.7.3. Prognostic Markers

Identifying biomarkers that predict recurrence, malignancy risk, or response to treatment is critical for patient stratification and long-term management. Longitudinal studies to assess outcomes in benign and malignant spiradenoma cases could provide data for evidence-based guidelines [33].

#### 3.7.4. Epidemiological Studies

Comprehensive studies to understand the incidence, demographic distribution, and risk factors of spiradenoma, including its malignant potential, are needed [33]. This information could guide early detection efforts and preventative strategies. Establishing comprehensive case registries and pooling data from multiple institutions can provide larger datasets for analysis. Reviewing existing cases across institutions can highlight trends in diagnosis, management, and outcomes.

#### 3.7.5. Exploration of Imaging Modalities

Further evaluation of advanced imaging techniques, such as MRI and PET/CT, could refine preoperative planning and help differentiate spiradenoma from other adnexal tumors or malignant lesions.

#### 3.7.6. Histological and Molecular Characterization

A deeper understanding of the histopathological features and molecular profile of spiradenoma and its variants will enhance diagnostic accuracy and the ability to predict malignant transformation. Even small sample sizes can yield significant insights when using advanced genomic and molecular techniques.

#### 3.7.7. Patient-Derived Models

Developing patient-derived organoids or xenograft models could help explore therapeutic options and tumor biology.

## 4. Conclusions

Owing to its rarity, accurate diagnosis is crucial for proper management and identification of potential malignant transformation. Accurate diagnosis is crucial for appropriate management and prognosis. Spiradenomas have distinct histological features; however, in some cases, they may present with components that resemble other conditions, such as ACC. Misdiagnosis could lead to unnecessarily aggressive treatment because ACC has a more concerning prognosis with the potential for local invasion and high recurrence rates. In conclusion, with an accurate diagnosis and appropriate surgical excision, the prognosis for spiradenoma is generally excellent. However, long-term follow-up is advisable because of the rare possibility of malignant transformation.

## Figures and Tables

**Figure 1 diagnostics-15-00173-f001:**
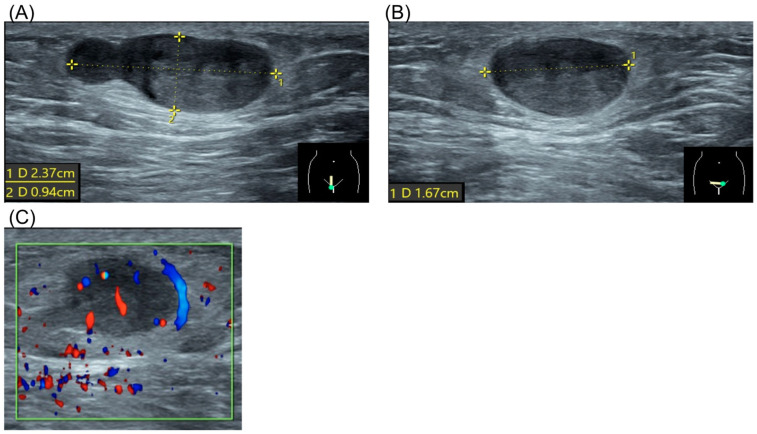
Ultrasound of spiradenoma. (**A**) Sagittal view of the tumor. Dotted line represented tumor diameter 2.37 × 0.94 cm. (**B**) Coronal view of the tumor. Dotted line represented tumor diameter 1.67 cm. (**C**) Doppler flow study of the tumor.

**Figure 2 diagnostics-15-00173-f002:**
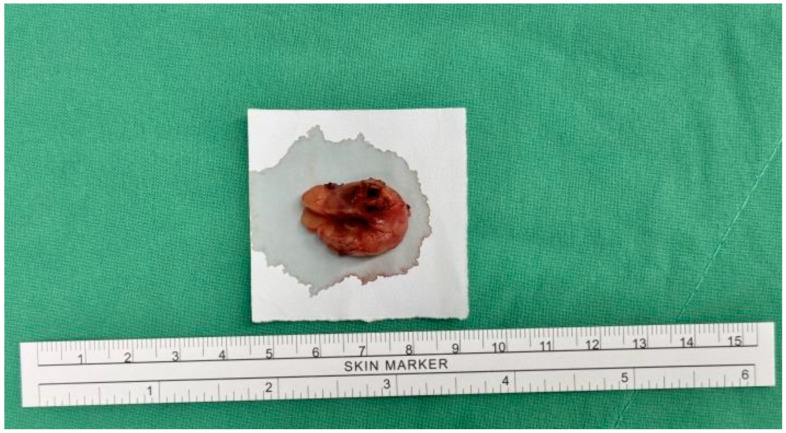
Gross appearance of the specimen.

**Figure 3 diagnostics-15-00173-f003:**
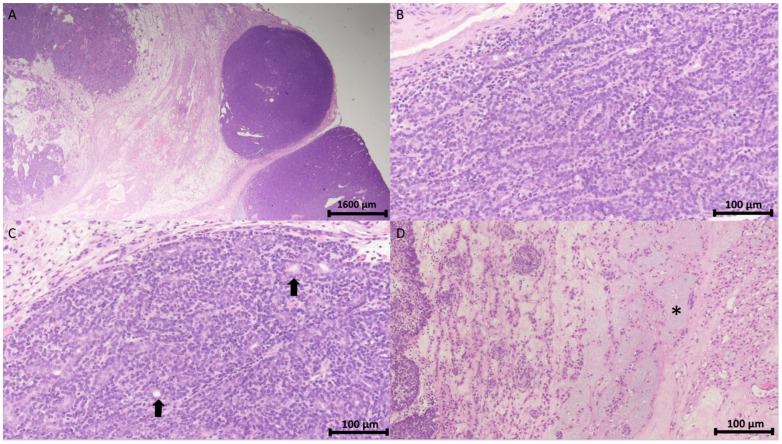
Histopathology of spiradenoma. (**A**) Pathology shows a well-demarcated encapsulated neoplasm with a multinodular pattern in the dermis and subcutis (H&E stain, 12.5×). Scale bar = 1600 μm. (**B**) The neoplasm comprises outer peripheral dark small basaloid cells and inner luminal light cells arranged in a trabecular pattern (H&E stain, 200×). (**C**) Cohesive fashion (H&E stain, 200×) with focal ductal differentiation (arrow). (**D**) Interconnected strands (H&E stain, 200×) and intercellular mucinous substance (asterisk). Scale bar in (**B**,**C**) = 100 μm.

**Figure 4 diagnostics-15-00173-f004:**
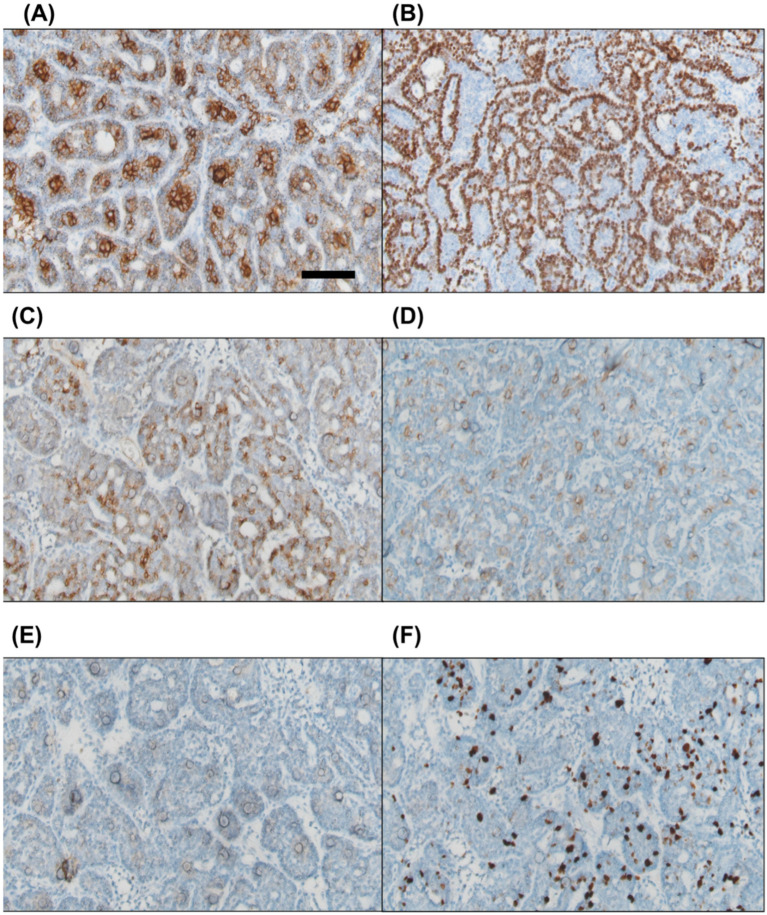
Immunohistochemical staining reveals that the inner luminal cells are positive for CK7 (**A**), while the outer basaloid cells show positivity for p63 (**B**). Focal positivity is observed for CD56 and CD117 ((**C**) and (**D**), respectively). PAX8 staining is negative (**E**). The Ki67 proliferation index is approximately 10–20% (**F**). Scale bar = 100 μm.

**Table 1 diagnostics-15-00173-t001:** Search strategy for the literature.

Items	Specification
Timeframe	From 2010 to 10 December 2024
Database	PubMed, Scopus, Web of Science, and Embase
Search terms used	“spiradenoma, skin, and adult”
Inclusion and exclusion criteria	All references were SCI-indexed articles written in English.
Selection process	Two independent reviewers evaluated the titles and abstracts to determine eligibility.

## Data Availability

All relevant data are reported in the article.

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
