# Peer review of "Spiradenoma: A Case Report and Review of the Literature"

_diagnostics, 2025, doi:10.3390/diagnostics15020173_

Round 1
Reviewer 1 Report
Comments and Suggestions for Authors
Dear authors
I found your article interesting. Please embedded in the context see my comments.

The language must be improved.
Author Response
Comment 1. please clarify “ in a 20-year case series”
Response 1: We thank the reviewer’s comment. We changed it to” D'Andrea et al. reported among 281 SAT within 20 years” (page 1, lines 28)
Comment 2: line 30: “as the most frequent benign”. It is in contrast with the first sentence of the first paragraph. Please clarify it.
Response 2: We thank the reviewer’s comment. We changed it to “spiradenoma was reported as the most frequent benign SAT (19.5%),” (page 1, line 29)
Comment 3. Line 57, Sagittal view of the tumor. Please add clinical figure of the lesion.
Response 3: We thank the reviewer’s comment. The lesion was a subcutaneous tumor. We cannot see the lesion over the skin. We did not have a picture of the skin.
Comment 4. line 61, “2-cm” Why did you general anesthesia instead of local one for such a simple excision?
Response 4: We thank the reviewer’s comment. We used general anesthesia to ease the patient’s tension.
Comment 5, line 63, “sutures” Please write if you consider margin for excision.
Response 5: We thank the reviewer’s comment. We have added “with a 1-cm safe margin”. (page 2, line 63)
Comment 6. line 75, Please add histochemical figures.
Response 6: We thank the reviewer’s comment. We have added the histochemical figure (Figure 4).
Comment 7. line 84. Please delete it.
Response 7: We thank the reviewer’s comment. We have deleted the sentence.
Comment 8, line 99, “including breast skin” Please list the body sites most frequently affected.
Response 8: We thank the reviewer’s comment. We have added the description. The statement reads as “the limbs are the most affected site”. (page 5, line 103)
Comment 9. line 110-111, Please clarify” Malignant SAT occupies 5.9%. They conclude SATs are rare and most benign”.
Response 9: We thank the reviewer’s comment. We have rewritten it. The statements read as “ Malignant SAT occupies 5.9% of cases. The authors conclude that SATs are rare overall, with the majority being benign.” (page 9 lines 115-116)
Comment 10. line 122. Please merge all sections. No need for sub-sectioning.
Response 10: We thank the reviewer’s comment. We have merged subsections 3.4-3.6.
Comment 11. line 138. Please mention that this is pathological feature “ and PAS-positive, hyalinized basement membrane material”.
Response 11: We thank the reviewer’s comment. We have added this is a pathological feature. The statement read as”pathological features show”. (page 6, line 142)
Comment 12. line 231, section. 3.9.1 Please cite references.
Response 12: We thank the reviewer’s comment. We have added a reference. (ref 32)
Comment 13. line 237 section. 3.9.2 Please cite references.
Response 13: We thank the reviewer’s comment. We have added a reference. (ref 33)
Comment 14. line 243 section. 3.9.3 Please cite references.
Response 14: We thank the reviewer’s comment. We have added a reference. (ref 34)
Comment 15. line 249 section. 3.9.4 Please cite references.
Response 15: We thank the reviewer’s comment. We have added a reference. (ref 34)
Reviewer 2 Report
Comments and Suggestions for Authors
1. Please have manuscript reviewed for scattered but numerous English grammar errors.
2. On page 3 change "Discussion" to "Literature Review and Discussion"
3. Omit the words "of the pubis" form the title. The case happened to be a spiroadenoma located on the pubis, but the literature review is of all spiroadenomas.
4. Explain: You conclude that more intensive studies of this disease are needed. How and why? How can you do such studies for a rare disease and why bother if it is indolent and much is already known.
Author Response
- Please have manuscript reviewed for scattered but numerous English grammar errors.
Response 1: We thank the reviewer’s comment. We sent our document for English editing.
- On page 3 change "Discussion" to "Literature Review and Discussion"
Response 2: We thank the reviewer’s comment. We have revised it accordingly. (page 4, section 3, line 89)
- Omit the words "of the pubis" form the title. The case happened to be a spiroadenoma located on the pubis, but the literature review is of all spiroadenomas.
Response 3: We thank the reviewer’s comment. We have deleted it accordingly. (page 1, title)
- Explain: You conclude that more intensive studies of this disease are needed. How and why? How can you do such studies for a rare disease and why bother if it is indolent and much is already known.
Response 4: Thank you for your insightful comment. While spiradenoma is a rare and often indolent tumor, we believe that further studies are warranted for several reasons:
- Understanding Rare Variants and Malignant Transformation: Although spiradenoma is generally benign, malignant transformation into spiradenocarcinoma, though rare, can have significant clinical consequences. Better understanding the molecular mechanisms and risk factors associated with this transformation is critical for improving patient outcomes.
- Genetic and Molecular Insights: Advances in genomic and proteomic techniques provide an opportunity to explore the genetic and molecular underpinnings of spiradenoma. This could yield insights not only for spiradenoma but also for related adnexal tumors, which share overlapping pathways and clinical behaviors.
- Challenges in Diagnosis: Spiradenoma can sometimes be misdiagnosed due to overlapping histological features with other adnexal tumors. More intensive studies could improve diagnostic accuracy, leading to more tailored treatment approaches.
- Therapeutic Implications: For rare cases requiring treatment (e.g., recurrent or painful lesions), understanding the tumor's biology might inform novel therapeutic strategies or minimally invasive interventions, which could benefit patients with similar adnexal tumors.
How Such Studies Can Be Conducted: Despite its rarity, meaningful research can be achieved through:
- Case Registries and Multi-Center Collaborations: Establishing comprehensive case registries and pooling data from multiple institutions can provide larger datasets for analysis.
- Retrospective Reviews: Reviewing existing cases across institutions can highlight trends in diagnosis, management, and outcomes.
- Molecular and Genomic Studies: Even small sample sizes can yield significant insights when leveraging advanced genomic and molecular techniques.
- Patient-Derived Models: Developing patient-derived organoids or xenograft models could help explore therapeutic options and tumor biology.
- Why It Matters: Although spiradenoma is indolent in most cases, understanding the spectrum of its clinical behavior, diagnostic challenges, and rare malignant potential remains essential for improving care. Rare diseases like spiradenoma can often serve as models for understanding broader principles in tumor biology and management, benefiting the field of dermatopathology and oncology as a whole.
We hope this addresses your concerns and provides a rationale for further study of spiradenoma. Several points we have added to the “section 3.9 Future direction and research needs”. (page 7, lines 247-250, page 8, lines 260-261)